# **Ensemble Random Forest for Tropical Cyclone Tracking**

Pradeebane Vaittinada Ayar<sup>1</sup>, Stella Bourdin<sup>2</sup>, Davide Faranda<sup>1,3,4</sup>, and Mathieu Vrac<sup>1</sup>

**Correspondence:** Pradeebane Vaittinada Ayar (pradeebane@laposte.net)

**Abstract.** Even though tropical cyclones (TCs) are well documented during the intense part of their lifecycle until they weaken, many physical and statistical properties governing them are not well captured by gridded reanalysis or simulated by Earth System Models. Thus, tracking TCs remains a matter of interest for investigating observed and simulated tropical cyclones. Two types of cyclone tracking schemes are available. On the one hand, some trackers rely on physical and dynamical properties of the TCs and user-prescribed thresholds, which make them rigid. They need numerous variables that are not always available in the models. On the other hand, trackers leaning on deep learning need, by nature, large amounts of data and computing power. Besides, given the number of physical variables required for the tracking, they can be prone to overfitting, which hinders their transferability to climate models. This study explores the ability of a Random Forest (RF) approach to track TCs with a limited number of aggregated variables. Our analysis focuses on the Eastern North Pacific and North Atlantic basins, for which 514 and 431 observed tropical cyclone track records are available from the IBTrACS database during the 1980-2021 period. For each 6-hourly time step, RF associates TC occurrence or absence (1 or 0) to atmospheric situations described by predictors extracted from the ERA5 reanalysis. Hence, the tracking is considered a binary supervised classification problem of TC-free (zero) and TC (one) situations. Then, situations with TC occurrences are stitched to reconstruct TC trajectories. Results show the good ability and performance of this method for tracking tropical cyclones over both basins and good temporal and spatial generalisation. RF has a similar TC detection rate as trackers based on TCs' properties and a significantly lower false alarm rate. RF allows us to detect TC situations for diverse predictor combinations, which brings more flexibility than threshold-based trackers. Last but not least, this study sheds light on the most relevant variables for tropical cyclone detection.

#### 1 Introduction

Tropical cyclones (TCs) are among the most devastating extreme events in terms of casualties and economic losses (Centre for Research on the Epidemiology of Disasters, 2021; UN Office for Disaster Risk Reduction, 2020). Several conditions are necessary for their formation. TC genesis requires warm sea surface temperatures to draw its energy from, low wind shear and ample humidity to ensure a stable vortex while maintaining the convection and adequate influence from the Coriolis force, combined with a pre-existing low-pressure disturbance in the atmosphere. Thus, a vortex is created around the depression

<sup>&</sup>lt;sup>1</sup>Laboratoire des Sciences du Climat et de l'Environnement, UMR 8212 CEA-CNRS-UVSQ, Université Paris-Saclay & IPSL, CEA Saclay, 91191, Gif-sur-Yvette, France

<sup>&</sup>lt;sup>2</sup>Atmospheric, Oceanic and Planetary Physics, Department of Physics, University of Oxford, Oxford, United Kingdom

<sup>&</sup>lt;sup>3</sup>London Mathematical Laboratory, 8 Margravine Gardens, London, W6 8RH, UK

<sup>&</sup>lt;sup>4</sup>Laboratoire de Météorologie Dynamique/IPSL, École Normale Supérieure, PSL Research University, Sorbonne Université, École Polytechnique, IP Paris, CNRS, Paris, 75005, France

with strengthening winds, and fuelled by ascending humid air (Emanuel, 2003; Weaver and Garner, 2023). It becomes a self-sufficient system that continuously draws energy from the ocean until reaching an unfavourable environment or land (the landfall). Then, the TC loses its energy, which causes its rapid dissipation (Kepert, 2010; Rüttgers et al., 2019).

Understanding how human-induced climate change influences TC activity remains a challenging scientific issue (Knutson et al., 2010; Walsh et al., 2016, 2019). Given the incomplete theoretical understanding of TCs and the limited observation period, studying the changes in their properties relies on model simulations (Knutson et al., 2019, 2020). Despite the tremendous effort made to increase the resolution of the Earth System Models (around 100 km for the last generation of models), it is still too low to simulate realistic TCs (Camargo and Wing, 2016; Roberts et al., 2020). Leveraging the recent advances in computational resources, a handful of global simulations with atmospheric spatial resolutions between 25 and 50 km are now available and reveal a clear improvement in simulating TCs (Murakami et al., 2015; Walsh et al., 2015; Roberts et al., 2020; Bourdin et al., 2024).

To study TCs simulated by global climate models, we need algorithms to detect them objectively. Such algorithms, known as TC trackers, are traditionally based on physical and dynamical properties of cyclones (see Zarzycki and Ullrich, 2017; Bourdin et al., 2022, and the reference therein for details about different trackers). These algorithms identify tropical cyclone points and connect them to reconstruct TC tracks employing thresholds applied to variables. Depending on the variables involved in the tracking process, Bourdin et al. (2022) defined two categories of trackers: physics-based and dynamics-based. Physics-based trackers rely on thermodynamic properties of a tropical cyclone, such as the local minimum sea-level pressure combined with a warm-core (temperature anomaly or geopotential thickness). Dynamics-based trackers rely on dynamical variables such as vorticity or other derivatives of the velocity. Both usually include an intensity criterion to discriminate between the systems.

35

The thresholds used in these trackers are tracking-scheme specific and subjective, and may also depend on the particular TC formation basin and the TC categories (Camargo and Zebiak, 2002; Befort et al., 2020). This may lead to a potential inability of tracking schemes to generalise to other domains or data from sources other than those used to calibrate the thresholds (Raavi and Walsh, 2020). To avoid subjective choice of thresholds and make the tracker more flexible in identifying cyclonic situations, the data-driven machine learning algorithms are the focus of this study. Indeed, these algorithms rely on data to identify cyclones based on different combinations of variables, independent of user-prescribed thresholds.

For instance, the detection skills of three machine learning approaches (Decision Trees, Random Forest, Support Vector Machines) and a model based on Linear Discriminant Analysis have been compared for satellite data in Kim et al. (2019a). Machine learning approaches showed better skill hit rates, while the linear approach showed lower false alarm rates. Among machine learning approaches, neural network-based deep learning approaches have lately gained attention for TC detection. Those are based on segmentation, edge detection, circle fitting, and comprehensive decision for satellite images (Kumler-Bonfanti et al., 2020; Wang et al., 2020; Nair et al., 2022). Kim et al. (2019b) leveraged a Convolutional Long Short-Term Memory network to detect and forecast hurricane trajectories on Community Atmospheric Model v5 simulation data.

However, these approaches use satellite and model data that can be biased and mainly focus on shape detection in images. As such, they are not comparable to the physics-based trackers previously mentioned, which have been developed from reanalysis and evaluated with respect to observed data and focus on TC-related physical processes. For instance, in Bourdin et al. (2022),

trackers were applied to the fifth generation of ECMWF Reanalysis (ERA5, Hersbach et al., 2020) and evaluated with respect to the observational record of the International Best Track Archive for Climate Stewardship (IBTrACS, Knapp et al., 2010). In that context, Gardoll and Boucher (2022) and Accarino et al. (2023) have developed convolutional neural network (CNN) architectures to detect cyclones. They used eight and six 6-hourly variables extracted from ERA5 in their CNN. The added value of such approaches is the ability to constrain the detection with the cyclone record provided by IBTrACS, by associating each set of 6-hourly data fields with the occurrence (absence or presence) of a cyclone (called labelling in machine learning). This makes tracking TCs a supervised classification problem.

The issue with using the latter type of algorithms in the case of TC detection is that the number of TC-related atmospheric situations is very low compared to TC-free situations. These algorithms trained with such data will learn from a larger diversity of TC-free situations and thus will be more accurate and inclined to rule for TC-free situations and, therefore, biased towards them. In addition, neural network-based algorithms need, by essence, large amounts of data, which can sometimes be qualified as data greedy. This calls for strategies to equilibrate the TC/TC-free ratio while keeping enough data to obtain a robust tuning of the CNN. Gardoll and Boucher (2022) reduced variable fields in the North Atlantic to  $8^{\circ} \times 8^{\circ}$  windows around the eye of the cyclone for every time step with a TC and sampled for each one of these windows two TC-free images, which drastically reduce the data sample (28,521 images). This potentially leads to overfitting and limits the generalizability of the tracker by lowering the diversity of TC-free situations and the spatial variability of the potential TC location due to the choice of windows around past TC locations. Only binary properties (TC/TC-free) of the tracker were evaluated in Gardoll and Boucher (2022). Accarino et al. (2023) considered non-overlapping 10°×10° windows over the whole joint North Pacific and Atlantic basins and opted for a data augmentation procedure of TC situations to reach a 50/50 ratio (425 358 images). Their CNN-based tracker produced comparable performance to the physics-based trackers in terms of TC track detection but generated larger numbers of false alarms, which is undesirable. Finally, this type of method processes large amounts of data, which calls for large computing power (typically GPUs in Gardoll and Boucher (2022) and a High Performance Computing infrastructure in Accarino et al. (2023)). Neither of the studies provided a physical interpretation of the tracker's performance.

In this study, the objective is to leverage and adapt a well-known and proven method, the Random Forest (RF, Breiman, 2001), to a TC tracking problem by associating a given atmospheric situation described by a limited set of predictors to the presence or the absence of TCs. This choice has been made by considering computing cost, the need for several meteorological variables, and the ultimate goal of such a tracker being the tracking of TCs in future climate simulations. Indeed, having many variables implies potential overfitting and impeded interpretation of the results and lower transferability to future climate simulations. Random forest provides interpretation means and lower computational costs. Higher data frugality will be achieved by considering simple variable statistics instead of entire variable fields, which will potentially improve the transferability of the tracking to climate simulations.

#### 2 Material and Method

## 2.1 Data

90

# 2.1.1 International Best Track Archive for Climate Stewardship, IBTrACS

Figure 1. a) Eastern North Pacific (ENP) and the North Atlantic (NATL) basins, with the TCs tracks and the associated wind intensity (in m  $s^{-1}$ ) used in this study. b) Boxes patching both basins. Only every second box is shown to improve clarity.

The IBTrACS "since 1980" set (Knapp et al., 2018) was retrieved in this study. In the following, two basins are going to be considered: the eastern North Pacific (ENP) and the North Atlantic (NATL) (cf. blue contours in Fig. 1a). The U.S. National Hurricane Centre (NHC) reports tropical cyclones' best tracks for these basins. First, extratropical cyclones are not considered in this study. Our study basins are limited to 30°N. Thus, only TCs are considered, and transitions to extratropical cyclones are not. For TC crossing this northward boundary, only the portion lying below 30°N is kept. The 42 cyclone seasons between 1980 and 2021 (from June to November in the Northern Hemisphere) are considered. At the time of this study, tracks in 2022 and later are removed, since some of them are still labelled provisional. Those labelled "spur", not providing maximum wind and minimum pressure, and not reaching the Tropical Storm (TS) stage, are removed. The TS stage is decided according to the storm category given by the values of the minimum sea level pressure  $P_{\min}$  and the 10-minute near-surface sustained wind  $u_{10}$ . Based on Table 2 of Bourdin et al. (2022), TS stage is reached when  $P_{\min} \leq 1005$  hPa and  $u_{10} \geq 16 \text{ms}^{-1}$ . Tropical cyclone

stage is reached when  $P_{\min} \le 990$  hPa and  $u_{10} \ge 29 \text{ms}^{-1}$ . Once processed, the ENP and NATL basins respectively contain 514 and 431 tracks at a 6-hourly timestep.

#### 105 2.1.2 ERA5



Our main objective is to associate climate variables and determine the main drivers that explain the formation and strengthening of TCs during their lifetime. Hourly estimates of atmospheric variables are available in ERA5 at  $0.25^{\circ} \times 0.25^{\circ}$  from 1979 to the present day. While having similar performances as JRA-55 or NCEP-CFSR for a range of metrics (Zarzycki et al., 2021; Roberts et al., 2020), ERA5 does not perform any specific assimilation for TCs (Zarzycki et al., 2021), motivating our choice to use ERA5 to evaluate the tracker developed in this paper. 6-hourly data from 1980 to 2021 are extracted, consistent with the period of the IBTrACS data. The choice of 6-hourly data stems from the overall objective to track TCs in climate model simulations, whose outputs are rarely at a higher temporal resolution. Five variables are extracted from ERA5:

- the mean sea level pressure, MSLP (in hPa),
- the 10-m wind intensity, UV10 (in  $m s^{-1}$ ),
- the total column water vapour, TCWV (in  $kg m^{-2}$ ),
  - the relative vorticity at 850 hPa pressure level, RV850 (s<sup>-1</sup>),
  - the geopotential thickness between 300 and 500 hPa pressure level, THZ300\_Z500 (in m).

These variables are described in Table S1 of the supplementary material. These variables have been selected based on their ability to characterise specific physical properties of TCs and their wide availability in climate model simulations' output. In particular, TCs have a warm core, with the most intense winds found close to the surface. TCs are structured with an eye at the centre, an eyewall, and spiral convective rain bands around them. TCs are driven by diabatic processes, meaning their energy comes from extracting oceanic moisture that releases latent heat once condensed in the upper troposphere. Considering this, MSLP characterises the spatially coherent low-pressure structure (the eye and the eyewall), UV10 the strong surface wind, TCWV the moisture and the potential for rain, RV850 the TC vortex and THZ300\_Z500 the upper-level warm core associated with the local depression in the TCs.

## 2.1.3 Data-set preparation

Several steps are followed to prepare the data. First, both basins are patched by  $20^{\circ} \times 10^{\circ}$  overlapping boxes (see shaded blue boxes in Fig. 1b), totalling 20 and 16 boxes respectively for ENP and NATL. This is done to deal with cases where two or more TCs occur at the same time in a given basin. Then, for every box, a vector of zeros and ones is constructed as follows: a box containing an IBTrACS point reaching TS intensity ( $P_{\min} \leq 1005 \text{ hPa}$  and  $u_{10} \geq 16 \text{ms}^{-1}$ ) is coded 1, and 0 otherwise at every timestep. Thus, the TC tracking problem is handled as a binary classification problem.

Then, ERA5 predictors associated with these binary vectors are built as follows: instead of considering the whole variable field within a box, only four statistics of that field are considered: minimum, mean, maximum and standard deviation. Thus, for a given timestep, the atmospheric situation within a box is described by a set of 20 predictors (5 climate variables × 4 statistics). Those predictors are labelled with the physical variable name attached to the statistic corresponding suffix (min, mean, max, sd). For instance, MSLPmin, MSLPmean, MSLPmax and MSLPsd are obtained for MSLP. Finally, for a given basin, the binary vector and the associated set of predictors of every box are concatenated and standardised (*i.e.* centred and divided by the standard deviation). Note that standardisation is not necessary for the current application of random forests. However, it has been made here to anticipate tracking TCs in climate models with biases compared to ERA5. The standardisation removes part of the mean and variance biases of the climate models and potentially eases the transferability to the tracker to climate models without recalibration. A table, with about 600 000 and 490 000 rows, is respectively obtained for ENP and NATL.

No formal test has been performed to demonstrate that using these four single-value statistics instead of the whole field in the box was better. Since only the presence or absence of a TC within one box, regardless of its position, is sought, these four statistics summarising the spatial structure are preferred to describe the whole  $20^{\circ} \times 10^{\circ}$  box. Furthermore, the ERA5 spatial resolution is  $0.25^{\circ}$ , resulting in 3200 grid-points per box for each physical variable. Using 16000 predictors to predict a single outcome does not seem reasonable.

## 2.2 Methods





# 2.2.1 Ensemble Random Forest for unbalanced data and experimental set-up

Random Forest (RF, Breiman, 2001; Hastie et al., 2009) is a supervised machine-learning algorithm based on generating an ensemble ("forest") of decision trees grown in parallel, referred to as bagging in machine learning. Each decision tree in the forest separates the target variable into homogeneous groups according to a sequence of *if-else* decision rules applied to the predictors. In our binary classification framework, each new separation according to a decision rule between the nodes has been performed via maximal impurity reduction, using the Gini index as an impurity function (Breiman et al., 1984). A random subset of data is provided for each tree (the in-the-bag dataset), and a random subset of covariates is tried at each node in each tree, bringing robustness to the classification. In this paper, such an implementation of RF is provided by the R package "ranger" (Wright and Ziegler, 2017) and follows the approach developed in Malley et al. (2012) to obtain the probabilities of a diagnosis of diabetes or appendicitis given sets of medical tests. Each classification tree gives a probability on the 0/1 class of a datum by taking the majority vote in a terminal node. The average probability of the trees is the RF probability estimate for class occurrence for each datum. A grid-search is performed on the three key parameters: (i) the number of trees, (ii) the random number of features considered to perform the best split to grow the trees and (iii) the minimal size of end nodes (not shown). Results showed that the impact of the hyperparameters is quite minimal, and no configuration of the hyperparameters yielded significantly better results. Therefore, the hyperparameters were set to the default values: 500 trees, 4 randomly chosen features to perform the best split and a minimal end node size of 10.

In the case of TC tracking, an imbalanced data problem presents itself. Indeed, the class "presence of TC" is underrepresented with only 2.1% (resp. 2.6%) of the data for NATL (resp. ENP). This results in low-performing RF due to two phenomena: (i) successive partitioning of the data when growing the decision trees causes them to 'see' fewer and fewer of the rarer class, thus fitting more and more to the majority class ("absence of TC"); and (ii) interactions between covariates can go unlearned by the decision trees due to the sparseness of the data induced by partitioning (He and Garcia, 2009). Kuhn (2013) discussed resampling methods that can resolve class imbalances, but there is little consensus on the best approach. Siders et al. (2020) compared different approaches and showed that combining the subsampling of the majority class with an ensemble of random forest (ERF), *i.e.*, the use of multiple random forests with different subsampling of that majority class, gave the best performance.

In this study, the ERF approach is leveraged to tackle the class imbalance issue. The subsampling of the majority class is performed by setting the number of zeros as n times the number of ones. Several setups are tested with  $n \in \{10, 15, 20, 25, 30, 35\}$  and referred to as 'n-times' setup, and one setup is referred to as 'FULL' without subsampling. To evaluate the effect of the subsampling, for each n, 100 RFs are performed with a different subset of zeros provided to each RF. Three experiments are set for each basin:

- 1. Calibration experiment: one training of the ERF is made using the whole data during the 1980-2021 period and validated over the same period, where all the tracks are sought to be reconstructed. It is only performed as a first-order evaluation of the tracker and its ability to reproduce the training data.
- 2. Validation experiment: a 6-fold cross-validation (see Fig. 2) where yellow years within each fold (35 years) are used to train the ERF. The validation is performed over tracks reconstructed for all the validation years (in blue) from the six folds, allowing for validating ERF over the whole 1980-2021 period. This cross-validation is chosen to minimise the effect of any potential trend and interannual variability in the TC statistics (frequency, intensity) and the changes in IBTrACS data quality. Most of the ERF evaluations will rely on this experiment.
- 3. Test experiment: from the training performed over the whole period in the calibration experiment for ENP (resp. NATL) basin, the TC tracks over the NATL (resp. ENP) are reconstructed over the same period. This is done to evaluate the generalizability of ERF.

Depending on the experiment, setup and basin, the training of one RF took between 1 and 10 minutes when performed on a laptop with an 11<sup>th</sup> Gen Intel®Core(TM) i7-1165G7 @ 2.80GHz with height cores and 16 Go RAM and between 30 seconds and less than three and a half minutes when performed on a computing node Intel®Xeon®CPU E5-2650 v2 @ 2.60GHz with 16 cores and 65 Go of RAM (8 Go would be sufficient).

# 2.2.2 Track reconstruction and matching






In a given box, if ERF gives a probability of TC above 0.5, the location of the TC is estimated by the position of the minimum of MSLP in that box. From there, tracks are reconstructed from one TC location to the next. 24-hour gaps within a radius of

**Figure 2.** Scheme of the 'Validation experiment': 6-fold cross-validation scheme over the 1980-2021. Yellow years are used for the calibration and blue for validation. One out of six years is used for validation, making seven out of 42 per fold.

450 km are allowed during the reconstruction. A track is kept only if it lasts at least 24h. Different thresholds below and above 0.5 have been tested (not shown). The result was (i) that the higher the threshold, the lower the ability to detect TC and (ii) that the lower the threshold, the higher the false alarms. This behaviour was quasi-linear, so we chose 0.5 to be performant in detecting while having a low false alarm rate. One can adapt this level according to the desired applications.

The track-matching procedure used in this study is similar to the one in (Bourdin et al., 2022). Let us consider, at time  $t_i$ , a point  $d_i$  of a detected track D. It is associated with the closest points of a given observed track O at each time  $t_i$  that is located closer than 300 km (with a possibility that such a point does not exist). In Bourdin et al. (2022), a sensitivity analysis was conducted on the 300 km distance limit in Appendix D. In a nutshell, it was shown that results are not sensitive to this limit, and 300 km was selected as a reasonable value. Points of O associated with any point of track D are denoted as  $O_D$ -paired. It is composed of NOD elements. There are four possibilities:

- 1. NOD=0: None of the points of D has been paired to a point in O, and D is considered to be a false alarm (FA),
- 2. NOD>0 and all the points in  $O_D$ -paired belong to the same observed track O: D is a match for O and considered a hit (Hit),
- 3. NOD>0 and all the points in  $O_D$ -paired belong to multiple observed tracks: D is a match for the observed track having the largest number of paired points and considered a hit (Hit),
  - 4. None of the points of a given O has been matched: O is a miss (Miss).

A final treatment is performed to complete the matching: if an observed track is paired with two or more detected tracks, these detected tracks are merged into a single track. It happens when parts of the same observed tracks are detected separately due to the filtering consisting of coding 0 every timestep in the observation IBTrACS that do not reach TS intensity.

#### 215 2.2.3 Evaluation metrics and analysis

200

205

The first evaluated aspect is the performance of ERF in terms of binary classification. For that, the Matthews correlation coefficient (MCC, Matthews, 1975) is used as a measure of the quality of binary (two-class) classifications. It has been introduced

by Yule (1912) and its values range from -1 to +1. A score of 1 represents a perfect prediction, 0 an average random prediction, and -1 an inverse prediction. The MCC is particularly useful when the classes are imbalanced, as it accounts for the imbalance in the calculation. It is similar to the Pearson correlation coefficient in its interpretation. The MCC is more informative than other metrics in evaluating binary classification because it takes into account the balance ratios of the four categories of the contingency (or confusion) matrix: true positives (TP), true negatives (TN), false positives (FP), false negatives (FN) (Chicco and Jurman, 2020). The MCC is computed from the confusion matrix (see Table A1):

$$\mathrm{MCC} = \frac{TP \times TN - FP \times FN}{\sqrt{(TP + FP)(TP + FN)(TN + FP)(TN + FN)}}.$$

220

The second aspect evaluated is the ability of ERF to reproduce observed TC tracks. Once all tracks are labelled Hit, Miss, and FA two detection skills metrics are defined, the Probability of Detection (POD, sometimes referred to as "Hit Rate") and the False Alarm Rate (FAR):  $POD = \frac{Hit}{Hit+Miss}$ ;  $FAR = \frac{FA}{FA+Hit}$ . POD and FAR are expressed in %, and good performance is achieved when POD is high and FAR is low.

Another aim of this paper is to provide some physical interpretation to the presence or absence of a TC given an atmospheric situation. Breiman (2001) proposed to evaluate the importance of a predictor variable (or feature)  $X_j$  for predicting Y (here the probability) by adding up the weighted impurity decreases  $p_t \Delta_i(s_t, t)$  for all nodes t where  $X_j$  is used, averaged over all trees  $\phi_m$  (for m = 1, ..., M) in the forest:

$$\operatorname{Importance}(X_j) = \frac{1}{M} \sum_{m=1}^{M} \sum_{t \in \phi_m} \mathbb{1}(j_t = j) \left[ p(t) \Delta_i(s_t, t) \right],$$

where p(t) is the proportion  $\frac{Nt}{N}$  of samples reaching node t,  $j_t$  denotes the identifier of the predictor used for splitting node t and  $\Delta_i(s_t,t)$  is the impurity decrease at split  $s_t$ . For each one of the 20 predictors, the feature importance is then the contribution in % of each variable to the total reduction of impurity.

Then the idea is to determine the importance of each predictor in the prediction of every single outcome (all zeros and ones) by RF. This is performed by computing the SHapley Additive exPlanation (or SHAP) values with the method proposed by Lundberg et al. (2020) with an implementation for tree-based algorithms provided in the R package "treeshap" (Komisarczyk et al., 2023). The general idea of SHAP values is to explain each outcome of RF as a sum of the effect  $\varphi_i$  of each predictor  $X_i$ . The SHAP value is  $\varphi_i$ , which stems from a concept introduced in cooperative game theory (Shapley, 1951). The idea is to determine the average contribution of a predictor to a specific prediction (here, a probability) to every combination of predictors. This can be written as follows:

$$\varphi_i = \frac{1}{\text{\# predictors}} \times \sum_{\substack{\text{combinations}\\ \text{excluding } X_i}} \frac{\text{marginal contribution of } X_i \text{ to combination}}{\text{\# combinations excluding } X_i \text{ of this size}}.$$

Once the SHAP values  $\varphi_i$  for all predictors  $X_i$  and for every outcome of the RF are computed, SHAP-based partial dependence plots are obtained by plotting  $\varphi_i$  against  $X_i$ . These plots will help to interpret the presence of TC given an atmospheric situation described by a set of predictors  $X_i$  and explore the evolution of TC probability according to the evolution of predictor  $X_i$ .

# 2.2.4 UZ algorithm

For comparison purposes, we use the UZ algorithm, a physics-based detection scheme developed in Zarzycki and Ullrich (2017) and implemented in TempestExtremes (Ullrich et al., 2021). It was shown in Bourdin et al. (2022) to have good detection scores with a particularly low False Alarm Rate. The UZ scheme relies on a 2-step procedure. The first step is the detection step to identify candidate TC points. These candidates are MSLP local minima associated with an upper-level warm core, which is measured by the geopotential thickness between 300 and 500 hPa pressure levels. The second step is the stitching step to link candidates and reconstruct tracks. The tracks must be associated with a maximum wind speed of at least 10 m/s over at least 54h. For more technical details, the reader is redirected to (Zarzycki and Ullrich, 2017; Ullrich et al., 2021; Bourdin et al., 2022).

#### 3 Results




# 3.1 Zero class subsampling choice

As mentioned in the method section 2.2.1, random forest is subject to being biased toward the majority (here, zero) class when applied to unbalanced data. In this section, the results of ERF for different subsampling over the NATL basin are used to select the best one. The MCCs for the validation experiment, given in the top panel of Figure 3, are quite similar for the different sub-samplings. It ranges from a little below 0.74 for the FULL setting to a little above 0.75 for the 20-times setting, with very little difference between the 15-times and 25-times settings.

POD and FAR metrics for the validation experiment are respectively given in the middle and bottom panels of Figure 3. The POD decreases almost linearly from 85% to 73% from the '10-times' to the 'FULL' setting. Similarly, the FAR also decreases from 18% to 5%. This indicates that a good ability to detect TCs goes along with a high level of generating false alarms. This also explains the similar MCC metrics for the different settings, indicating some compensation between the four categories of the confusion matrix. The subsampling '25-times' setup has the *medium* performance, with POD around 78% and FAR around 8% (see Fig. 3), is chosen.

The other result drawn from Figure 3 is that the effect of the sampling of zeros given an n-times setup on MCC, POD, and FAR is very marginal, considering the narrow boxplots. It means that even though the tracks reconstructed from the average probability obtained from the 100 RFs are used in the following of this study, a lower number of RFs would be sufficient.

#### 3.2 ERF detection analysis

Figure 4 shows four examples of TC tracks reconstructed over the NATL basin for the validation year 2017, from the average probability obtained from the 100 RFs with the '25-times' setup. Similar reconstructed tracks for the ENP basin with a similar ERF setup are shown in Figure S1 of the supplementary material. TC tracks reconstructed from ERA5 with ERF are very close to the observed tracks from IBTrACS, even though the trajectories have very different shapes. Note the long gap in the Harvey cyclone (Fig. 4a) is due to the filtering consisting of selecting only the time steps reaching TS intensities (see Sect. 2.1.1).

**Figure 3.** Validation experiment boxplots of MCC (*top*), POD (*middle*) and FAR (*bottom*) obtained over validation years obtained for the 100 RFs of different ERF with different subsampling of non-TC situations (*i.e.* zeros), for the NATL basin. The *top* and *bottom* fences are situated at 1.5 times the interquartile range from the box, and the dots are the values beyond these fences. The orange line represents the median value. Violet symbols represent the metrics for the tracks obtained from the average of the probabilities given by the 100 RFs of ERF.

Table 1 gives the POD and FAR metrics for the tracks reconstructed from the average probability from ERF for calibration, validation, and test experiments for the '25-times' setup for both basins. For the validation experiment, POD are respectively 77.5% and 77.8% for ENP and NATL basins. FAR are respectively 8.7% and 7.9% for ENP and NATL basins. For calibration experiments, POD are above 90% and FAR is around 2% for both basins. Note that the choice of validation data (one year every 6 years) in the cross-validation scheme in the study was only one possibility among others. A test (not shown), with a 6-fold cross-validation scheme in which validation years are stacked (1980-1986,…,2015-2021), yielded similar results for the validation experiment, and results are even better when only the last fold, with the latest validation years (2015-2021), is considered.



**Figure 4.** ERF-based TC tracks reconstructed over validation year 2017 for the NATL for the '25-times' setup. a) Harvey, b) Irma, c) Jose and d) Maria.

**Table 1.** POD and FAR for tracks reconstructed from average probability from ERF for calibration, validation, and test experiments for the '25-times' setup and for UZ in %. Multi-basin refers to POD and FAR from ERF applied to both ENP and NATL basins under the '25-times' setup discussed in Sect. 4.1. The right part of the table, referred to as 'Ablation experiments', gives the POD and FAR for ERF experiments conducted with a reduced number of predictors discussed in Sect. 4.2

|             | Main experiments |      |             |      | Ablation experiments |      |              |      |
|-------------|------------------|------|-------------|------|----------------------|------|--------------|------|
|             | ENP              |      | NATL        |      | ENP                  |      | NATL         |      |
|             | POD              | FAR  | POD         | FAR  | POD                  | FAR  | POD          | FAR  |
| Calibration | 91.1             | 2.0  | 93.6        | 2.3  | 89.7                 | 4.1  | 89.5         | 3.6  |
| Validation  | 77.5             | 8.7  | 77.8        | 7.9  | 77.2                 | 11.9 | 76.7         | 13.2 |
| UZ          | 76.4             | 24.1 | 78.4        | 15.0 | -                    | -    | -            | -    |
| Test        | NATL (calib.)    |      | ENP (calib) |      | NATL (calib)         |      | ENP (calib.) |      |
|             | 76.6             | 15   | 68.4        | 7.8  | 73.5                 | 15.5 | 69.3         | 15.5 |
|             | Multi-basin      |      |             |      |                      |      |              |      |
| Calibration | 91.8             | 3.1  | 90.9        | 2.4  | 90.5                 | 5.5  | 87.8         | 3.1  |
| Validation  | 79.2             | 9    | 74.8        | 5.3  | 78.1                 | 13.7 | 70.1         | 8.3  |

In the following, the statistical and physical properties of the detected tracks are investigated. Figure 5 a) shows the track duration histograms for the observed, ERF-detected tracks (Hit and FA) and missed tracks. ERF-based Hits have a duration

**Figure 5.** Statistical properties of IBTrACS (purple), ERF-detected TC tracks: Hit (blue) and FA (red) and ERF-missed tracks: Miss (green). a) TC tacks duration histograms, b) Boxplot of ERF-based average probabilities associated with each time step of Hit, FA, and Miss tracks.

distribution quite similar to IBTrACS tracks, but with substantial differences for short-duration tracks (1 to 3 days). These short-duration tracks have a typically short lifespan and are of lower intensity. This discrepancy is supported by the duration distribution of Miss tracks, which is mainly short-duration tracks (the majority of them last between 2 and 4 days). False alarms are also of the same duration. Differences in probabilities of TCs given by ERF associated with every time step of Hit, FA, and Miss of tracks are then investigated. Figure 5b shows TC probabilities conditionally on its labelling as Hit, Miss, or FA. Probabilities associated with Hit tracks (median above 0.9) are substantially different compared to those associated with FA (median a little above 0.6). This means that even if FA tracks are detected (probability >0.5) by ERF, FA are less likely to happen than Hits. Miss tracks are associated with very low probabilities, meaning they are completely missed by ERF while having been recorded in IBTrACS.

To investigate how these different tracks diverge in nature, the maximum wind and minimum sea level pressure associated with these different types of tracks are considered. Figure 6c shows the scatter plot of maximum surface wind against minimum sea level pressure associated with every timestep of observed, detected (Hit and FA) and missed tracks. Figure 6 a and b, respectively, give the associated maximum wind and sea level pressure histograms. In general, and as already pointed out in Bourdin et al. (2022) and Dulac et al. (2024), the wind-pressure relationship in ERA5 is different from the one in the observations (purple dots versus the rest). Detected TCs are weaker than observed ones. In particular, Hit tracks barely reach category 4 when considering ERA5 minimum sea level pressure, and it is even worse when considering ERA5 maximum wind: Hit tracks barely reach category 3. In addition, these figures also provide insight into the FA and Miss tracks. Miss tracks are, for the majority of them, associated in ERA5 with minimum pressure above 1005 hPa and maximum wind below 16 m s<sup>-1</sup>, which are the TS intensity threshold. It means that these tracks are missed because ERA5 is failing to represent these TCs correctly. Concerning FA tracks when examining Figure 6 a-c, the maximum winds and minimum pressure are located around 16 m s<sup>-1</sup> and 1005 hPa pressure, which are again the threshold for TS intensity. Thus, these FA tracks may be related to the uncertainty of ERF, which associates an atmospheric situation with a TC even though none has been observed.

**Figure 6.** Physical properties of IBTrACS (purple), ERF-detected TC tracks: Hit (blue) and FA (red) and ERF-missed tracks: Miss (green). a) histograms of maximum surface wind [in m s $^{-1}$ ], b) histograms of minimum sea level pressure [in hPa] and c) the scatter plot of maximum surface wind against minimum sea level pressure. Vertical lines indicate the TC intensity classification Saffir-Simpson Hurricane Scale thresholds of 10-minute sustained wind. Horizontal lines indicate pressure thresholds based on Klotzbach et al. (2020).

Figure 7 shows Miss and FA tracks distributed over the NATL basin and the associated ERF-based average probability, the minimum pressure and the maximum wind. Miss tracks are distributed over the entire domain and confirm the results of low probability and intensity in Figures 5 and 6. However, one track shows high probability, intense wind and low pressure pictured in reddish colours in the three left-hand side panels of Fig. 7. This track illustrates one drawback of dividing the basin into  $20^{\circ} \times 10^{\circ}$  overlapping boxes: ERF can only detect one TC at a time within a box. However, two TCs may happen at the same time within one box. Figure 8 shows the IBTrACS track of the TC IRIS spotted in Figure 7 and the stronger TC HUGO occurring at the same time. The probability, the pressure, and the wind associated with the missed TC IRIS in Figure 7 are those of the strong TC HUGO. The FA tracks are mostly distributed at the edge of the domain. In particular, they are located in areas where TCs are typically weaker: the primary development region (eastern part of the domain between  $10^{\circ}$ N and  $20^{\circ}$ N) where TCs are developing, and coastal areas where they disappear. This can be related to the uncertain aspect of these tracks that are yielded by lower probabilities.



**Figure 7.** Average probability (top row), minimum sea level pressure, Pmin [in hPa] and maximum wind U10max [in m s<sup>-1</sup>] for Miss tracks (left column) and FA tracks (right column). Colours are saturated for Pmin and U10max.

Similar figures for the ENP basin are given in Figures S2 to S5 of the supplementary material and give similar conclusions as for the NATL basin. The major difference is that the distinction in terms of intensity between Hit and FA is less obvious based on ERA5. The wind-pressure relationship in ERA5 compared to the observation is even worse for ENP, where TCs barely reach category 2 intensity for the wind and pressure scale. The median probability of Hit and FA tracks is closer (0.8 vs 0.65) and yields a higher FAR ratio. Even though the majority of FA tracks are associated with wind and pressure around 16 m s<sup>-1</sup> and 1005 hPa, some of them present more intense values. One hypothesis may be that these tracks have not been recorded in IBTrACS.


Figure 8. Example of TC IRIS that has been missed by ERF due to the presence of a stronger TC HUGO and the associated Pmin [in hPa].

**Figure 9.** Boxplot of Gini-based feature importances from the 100 RFs of the ERF for the calibration experiment and the '25-times' setup for a) ENP and b) NATL.

#### 3.3 Physical interpretation

In this section, the contribution of the different predictors to the detection of TCs is analysed to provide physical insights into the presence or absence of a TC. Figure 9 shows that for both basins, the six variables with the largest feature importance are the same: RV850sd, MSLPmin, UV10max, RV850max, THz300\_z500max and TCWVmax. These predictors are physically well-founded in explaining the presence of a cyclone. RV850sd characterises the singularity of the vortex within a box: the higher it is, the more the TC vorticity stands out from the vorticity of the rest of the area within a box. It is more important than the RV850max, the fourth most important variable. Then, UV10max and MSLPmin are the following most important variables. This makes sense, since they are associated with the strong surface winds and the location of the cyclone's eye,

respectively. The following variables are the TCWVmax and THz300\_z500max. The former reflects the potential for rain and the moisture lifted by the TC; the latter characterises the upper-level warm core associated with the TC. Note that the order of importance is slightly different between the basins. For instance, maximum wind is more important than sea level pressure for the NATL basin, while it is the opposite for the ENP basin. It may result from the different wind-pressure relationships between both basins (see Fig. 6c and Fig. S3c of the supplementary). TCWVmax is less important in explaining the presence of the TC situation for the ENP basin.

Feature importance quantifies the average contribution of a given predictor in discriminating TC from non-TC situations. However, it would be interesting to determine the contribution of each predictor to each outcome of an RF. Indeed, we want to evaluate the ability of RFs to learn the relevance of each predictor to drive each TC/non-TC prediction. This is provided by the SHAP-based partial dependency plots shown in Figure 10 for the NATL basin. This figure pictures the relationship between the six top predictors (according to feature importance) and their respective SHAP values. Note that given the computing time, SHAP values are computed for only one RF among the 100 RFs of the calibration experiment and the '25-times' setup. These partial dependence plots are probably very similar for the 100 RFs, given the small dispersion of the MCC, POD and FAR performances' metrics (see Fig. 3) and feature importance (see Fig. 9). Let us consider the partial dependency plot in panel a) of Figure 10. On the abscissa is given the physical range of RV850sd and the associated SHAP values on the ordinate. It shows the contribution of RV850sd given its value to the probability value of TC occurrence.

The partial dependence is distinct between the "zero" and the "one" populations, with marginal overlap. For the zeros, the SHAP values are always very close to 0, while for the ones, the SHAP values always steeply increase when the associated predictor increases (except MSLPmin, SHAP values increase when it decreases). For these six predictors, SHAP values tend to reach a cap value after the predictors reach a certain level, and they even decrease for MSLPmin and UV10max. This means that the contribution of these predictors in discriminating TC from non-TC situations does not change when reaching an intense value. This figure also shows that TC situations can occur for a large range of values and diverse combinations of these predictor values. This advocates for TC tracking approaches that bring more flexibility than threshold-based approaches. Similar results can be drawn based on Figure S6 of the supplementary material for the ENP basin. This analysis highlights the advantages of using random forest over traditional trackers that rely on the sequential use of thresholds, as most variables exhibit nonlinear interdependencies.

# 3.4 Comparison with UZ






POD and FAR for UZ in both basins are reported in Table 1. POD are close to those of ERF (Validation experiment) with 1% lower POD for ENP (76.4%) and less than 1% higher (78.4%) for NATL. However, FAR, which reaches 24.1% in the ENP and 15% in the NATL, amounts to almost three and two times the ERF scores, which is undesirable.

Figures 5 to 7 have been reproduced for UZ and both basins. They have been added to the supplementary material as Figures S7 to S12. Panel b of Figure 5 and the two top panels of Figure 7, showing the probability associated with the tracks, are irrelevant for UZ. For both basins, the track properties obtained from UZ are similar to those obtained with ERF (see Figs. S7, S8, S10 and S11). The main difference resides in the properties of missed tracks. These tracks are similarly short but more

**Figure 10.** Partial dependence plot for top six predictors a)-f) obtained for one of the 100 RFs of the calibration experiment and the '25-times' setup for the NATL basin. Contour lines indicate the density of the scatter plot between one predictor and the associated SHAP values. Yellow and blue characterise, respectively, the density of the zeros (probability<0.5) and the ones (probability>0.5) population. Vertical and horizontal lines, respectively, indicate the median of the predictors and the associated SHAP values for both populations. The distributions of the predictors and SHAP values are also given conditionally to both populations.

frequent and slightly more intense (higher maximum wind and lower SLP). Properties of FAs are similar for UZ and ERF in the NATL basin. In the ENP basin, FAs from UZ are significantly more frequent and intense than the FAs generated by ERF. The spatial distribution of Miss and FA of the UZ tracker is also similar to that of ERF (Fig. S9 and S12). Miss tracks are distributed over the entire domain with low intensity, and FA tracks are mostly located in the primary development region and coastal areas, where TCs are typically weaker.

To further explore the similarities and differences between UZ and ERF, Figure 11 shows the number of tracks that are common for observations (IBTrACS, IB) and tracks that are detected by ERF (Validation experiment and '25-times' setup) and UZ. A large portion of the tracks of IBTrACS are detected by both UZ and ERF (335/252 for ENP/NATL basins), and some tracks are only detected by UZ (28/31 for ENP/NATL basins) or ERF (27/22 for ENP/NATL basins). In total, the Hit numbers are similar in both methods. UZ produces more FAs (115/50 for ENP/NATL basins) than ERF (41/31 for ENP/NATL basins). Finally, there are no common FAs between ERF and UZ. This means the FA appear for different reasons in the two methods.

**Figure 11.** Number of tracks in common between observed tracks (IBTrACS, IB) and detected tracks from ERF and UZ for the following case: all three datasets, two out of the three and tracks specific to a dataset for the Validation experiment and '25-times' setup. a) ENP and b) NATL basins.

# 4 Sensitivity test


# 4.1 Need for regional tracker

In this study, ERF has been applied separately for each basin, which is uncommon in the literature. Usually, the tracking algorithm is applied over multiple basins (Bourdin et al., 2022; Accarino et al., 2023). This has been done to test the spatial generalizability of the ERF approach. This ability of ERF is based on the test experiment, which consists of reconstructing the tracks of the ENP basin using the ERF fitted for the calibration experiment ('25-times') setup for the NATL basin and vice versa (see sect. 2.2.1). Table 1 reports the FAR and POD for these test experiments. POD and FAR are respectively 76.6% and 15% for the ENP basin and 68.4% and 7.8% for the NATL. In the case of ENP, the test POD is similar to the validation one, while the FAR is degraded. It is the opposite for NATL. This shows a certain specificity of the TC tracking according

to the basin, which may stem from different factors. For instance, the latitudinal distribution of the cyclones is quite different between both basins: for the ENP basin, TC tracks are mostly located between 10°N and 20°N (see Fig. 1) while they are distributed all over the basin for NATL. Thus, this may involve different processes between TC in NATL and ENP. Differences in feature importance in Figure 9 between the two basins may be an illustration of that. This can also be due to the differences in the quality of TC representation between the two basins in ERA5. For example, differences in the wind-pressure relationship between both basins illustrate this quality difference (see Fig. 6c and Fig. S3c).

To highlight the need for a regional tracker, ERF has been carried out under the '25-times' setup for the data pooled from both basins, referred to as 'Multi-basin'. Table 1 gives the POD and FAR for calibration and validation experiments. Compared to basin-specific experiments, the performances are close, with slightly better POD and slightly worse FAR for the ENP basin. In contrast, it is the opposite for the NATL basin, with larger differences. The better performance for the ENP basin can be explained by its higher weight on the ERF training, given its larger data size (see Sect. 2.1.3) and the small total number of TCs. The results remain, nevertheless, better than UZ (see Tab. 1). It is therefore up to the user to decide if either one ERF for all basins or one specific ERF for each basin is necessary by considering if the loss of performance of the Multi-basin ERF compared to the regional ERF is acceptable or not.

# 405 4.2 Ablation experiment




An ablation experiment is conducted to get a more parsimonious ERF by reducing the number of predictors. Based on the feature importance (Fig. 9), the top six predictors are kept (see Sect. 3.3). Such a model is expected to generalise better, have a smoother behaviour when looking at partial dependencies, and be potentially more intrinsically interpretable.

In this section, the Calibration and Validation (for regional and Multi-basin) and Test (for regional) experiments have been performed under the '25-times' setup. The right part of Tab. 1 gives the POD and FAR from all experiments carried out with a reduced number of predictors. For all experiments, PODs are only slightly reduced but remain very close to the POD without the ablation. Interestingly, FARs are more strongly degraded (sometimes doubled) with the ablation. This means that predictors with lower feature importance control FA, indicating that we ought to be cautious when removing predictors. SHAP-based partial dependency plots for the validation experiment are given in Figures S13 and S14 for the Validation experiment and both basins. In general, these figures are similar to those of ERF performed with the full set of predictors. The only difference is the better distinction between "zero" and "one" populations, which can result from the higher FA rate.

To explore how many variables can be removed without deteriorating the performance of the tracker, a recursive feature elimination on the 6-fold cross-validation set-up (validation experiment) has been performed. It consists of eliminating the least important variable at each iteration. The results (not shown here) show that the performance of ERF remains stable until we remove 12 to 14 variables, which roughly corresponds to the number of variables we kept for the sensitivity test. The exception resides in FAR, which shows a 50% increase when we remove no more than 6 variables, confirming the role of the least important in controlling the false alarms.

#### 5 Summary and perspectives







In this study, we used random forest for tracking tropical cyclones in the eastern North Pacific and North Atlantic basins over the 1980-2021 period by associating atmospheric situations described by five climatic predictors extracted from ERA5 with observational IBTrACS records of tropical cyclones. More precisely, the tracking problem in this paper was equivalent to performing binary classification over imbalanced data containing substantially more TC-free situations than TC. This imbalance problem was addressed by combining an ensemble of random forests with the subsampling of TC-free situations. Before applying this method, the amount of data fed to it was reduced by considering four statistics of each predictor instead of its whole field (minimum, mean, maximum and standard deviation). In addition, basins were patched by overlapping boxes. In this way, our approach was able to learn the characteristics related to the presence of the TC inside a box, regardless of its position. It allowed us to detect cyclones that occur simultaneously within one basin or in both basins (for the multi-basin experiments).

Our data-driven ERF tracker showed good performances for detecting TC tracks: In validation, POD/FAR of 77.5%/8.7% and 77.8%/7.9% were obtained for the ENP and the NATL basins, respectively. Compared to the physics-based UZ tracker, used as a benchmark in this study, ERF showed similar POD but better (*i.e.*, lower) FAR. UZ was chosen because it was the most accurate among the physics-based trackers (Bourdin et al., 2022), and it was also better than other data-driven trackers (*e.g.*, the deep learning approach in Accarino et al., 2023). ERF has the advantage of requiring low computing power (see Sect. 2.2.1). Tracks detected by ERF have similar duration frequencies to IBTrACS tracks, except for short (2 to 4 days) and lower intensity tracks (see Figs. 5 and 6). Missed and false alarm tracks are mainly short tracks (1 to 3 days). Detected TCs have weaker intensity in ERA5 than in IBTrACS, due to ERA5 systematically underestimating TC intensity. So much so that it is likely that some cyclones are missing because they were reanalysed as too weak to be detected. False alarm tracks are very close to the tropical storm intensity thresholds and thus illustrate the uncertainties of ERF. These tracks are located in developing and landfall areas of cyclones, where their signal is less clear and more uncertain.

For both basins, the six most important variables for detecting the presence of TCs are the same and characterise the main physical and thermodynamic properties of TCs. The order of importance differs between the basins, highlighting potential specificities in the TC patterns and processes depending on the basin. Relying on the SHAP-based partial dependency plots, we showed that TCs can be detected through potentially diverse combinations of predictor values. This brings more flexibility than physics-based approaches that need user-prescribed thresholds.

Two aspects of our ERF trackers were then tested: the spatial generalizability of ERF, and the possibility of reducing the number of predictors. When the tracking was performed in one basin based on an ERF calibration performed on the other basin (resp. on both basins): (i) for ENP, the POD is similar (resp. improved) and the FAR is degraded (resp. similar), and (ii) for NATL, the POD is degraded (resp. degraded) and the FAR is similar (resp. improved). This shows an overall good ability for spatial generalizability of ERF, while showing potential need for regional tracking that can stem from the specificities of TC tracks and the differences in ERA5 quality between both basins. The ablation experiment showed that reducing the number of predictors according to their feature importance does not change (or only very marginally) the POD, but strongly degrades the FAR for all experiments. This showed a certain control over false alarms of the removed predictors.

As future work, the next focus will be on extending the tracking for all the basins at a global scale. Some tests will also be necessary to choose the minimum number of RF (lower than 100) and to determine which predictors control FAs and which predictors to remove without degrading the performance of the tracker. From there, the major challenge will be to apply ERF to Earth System Models from the Coupled Model Intercomparison Project phase 6 (CMIP6, Eyring et al., 2016) and, in particular, to the subset of CMIP6 simulations from the High Resolution Model Intercomparison Project (HighResMIP, Haarsma et al., 2016). Indeed, HighResMIP simulations are better at simulating TCs, and their tracking has already been done with other physics-based trackers (Roberts et al., 2020).

The primary target will be to apply the ERF tracker calibrated from ERA5 directly to HighResMIP simulations without any new ERF calibration. This raises the question of the transferability of the ERF calibrated from ERA5 to the models. This issue is two-fold: first, is there a need for bias correction? And second, will ERF be transferable to future climate projections given the climate change signal? The first question will deal with the possibility of a mismatch between the models and ERA5, preventing ERF from detecting cyclones. The second one will address the possibility that climate change will induce nonstationarity strong enough to prevent ERF from detecting cyclones.

Multiple applications can be foreseen. For instance, we will study the statistical and physical properties of TCs detected under climate change. More precisely, we will be able to compare them to the cyclones detected by physical-based trackers and evaluate the complementary added value brought by the flexibility of ERF for detecting cyclones. Furthermore, the differences in the relation between the predictors and the TC presence probability for the models and ERA5 will be evaluated using the partial dependency plots. Another application can be dedicated to climate change attribution studies by comparing the properties of TCs in simulations realised under controlled emission scenarios and future climate scenarios.

Ultimately, we would like to make our method widely available. Hence, efforts will be made to make it easy to use through open-source software.

Data availability. ERA5 data are available on the Copernicus Climate Change Service Climate Data Store (CDS, https://cds.climate.copernicus.eu/datasets/reanalysis-era5-pressure-levels?tab=download, last access: April 2024). The IBTrACS database is provided by NOAA, National Centres for Environmental Information, https://www.ncei.noaa.gov/products/international-best-track-archive (last access: April 2024).

# **Appendix A: Tables**




Author contributions. PV conceived the study, prepared the data and figures, conducted the analysis and wrote the original manuscript. SB contributed to the comparison with the UZ approach. MV discussed the experimental setup. All authors discussed, commented and edited the manuscript.

Table A1. Confusion Matrix

|               |   | Prediction |    |  |
|---------------|---|------------|----|--|
|               |   | 0          | 1  |  |
| Observations  | 0 | TN         | FP |  |
| Coser various | 1 | FN         | TP |  |

true positives (TP), true negatives (TN), false positives (FP), false negatives (FN)

*Competing interests.* The authors have no competing interests to declare.

Acknowledgements. Pradeebane Vaittinada Ayar thanks Tom Beucler of the University of Lausanne for his comments. All computations and figures are made using the R free software (R Core Team, 2024). This work has mainly received support from the European Union's Horizon 2020 research and innovation program within the project XAIDA: Extreme Events – Artificial Intelligence for Detection and Attribution (Grant agreement No. 101003469).

This work also benefited from state aid managed by the National Research Agency under France 2030, bearing the reference ANR-22-EXTR-0005 (TRACCS-PC4-EXTENDING project). The authors acknowledge the support of the INSU-CNRS-LEFE-MANU grants (projects CROIRE and COESION), and of the Institut Pascal at Université Paris-Saclay with the support of the program TROPICANA, under the reference ANR-11-IDEX-0003-01.

SB received financial support from the NERC-NSF research grant n° NE/W009587/1 (NERC) & AGS-2244917 (NSF) HUrricane Risk

Amplification and Changing North Atlantic Natural disasters (Huracan), and from the EUR IPSL-Climate Graduate School through the
ICOCYCLONES2 project, managed by the ANR under the "Investissements d'avenir" programme with the reference 37 ANR-11-IDEX0004-17-EURE-0006.

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
