# Peer review of "Ensemble Random Forest for Tropical Cyclone Tracking"

_EGUsphere, 2025_

## Author Comment (AC1)

"Ensemble Random Forest for Tropical Cyclone Tracking" by

P. Vaittinada Ayar et al.

We first would like to thank the anonymous reviewer for her/his thorough reading and very positive and constructive comments. We tried to take them into account as much as possible. A detailed point-by-point reply to these comments is provided below. Changes in the manuscript are indicated in **blue**.

**Answer to Referee #1**

**Overview**

This work applies Random Forest (RF) models to track tropical cyclones using environmental variables from a global reanalysis (ERA5) with an eventual goal of using the RF tracker in long-running climate simulations. The Eastern Pacific and Northern Atlantic TC basics were chosen for investigation. Random Forests were trained by categorising localised boxed regions in each basin as either containing a TC or not (TC-free) and associating statistics of environmental variables in each box from ERA5 to the binary events. Variables of mean sea level pressure, relative vorticity, column water vapour, and thickness were used as they represented different facets of physical mechanisms and TCs. Statistics are computed for these variables and included as inputs during RF training.

Training is conducted with 6-fold cross-validation to generate a range of RF solutions that are then used to compute MCC, POD, and FAR over a series of subsampling experiments – the authors note a significant proportion of their samples are TC-free compared to TC samples. Generally, a ratio of 25-1 is seen as reasonable with POD and FAR tradeoffs as the ratio is increased/decreased. Detection skill is notably better than the baseline UZ method in both basins. Further investigation of skill suggests the model primarily misses TCs at low intensity and low duration. The authors also devise analyses to interpret physical meaning, although I have some comments on this aspect of the analysis below.

Overall, the authors have employed RFs in a very unique and potentially innovative application area to track TCs in global reanalyses. The manuscript could benefit from improved grammar and clarity in locations, along with consideration of additional analyses or methods to improve the scientific presentation. I look forward to seeing a revised manuscript after careful revision.

**Comments**

✳*Comment– Lines 119-125 : If one of the objectives of the manuscript is to determine the physical relationships that govern TC tracks, why not allow the ML model to do the variable selection for you? Provide X number of variables and employ variable selection procedures like sequential forward selection or sequential backwards selection? Or use explainable AI techniques to do variable selection? If we are informing which variables the RF should learn from, aren't we prescribing our own biases into the physical mechanisms?*

**RESPONSE–** In our study, we provided 20 variables (five physical variables and the four associated statistics, see lines 139-145), that are preselected based on physical considerations mentioned in lines 126-132 and are based on past literature. Regardless of the number of predictors, we are not informing RF which predictors it should learn from. RF is expected to make the difference between the informative variables and those that are not for the considered problem, namely, the TC probability estimation. Therefore, even if we were to prescribe our own biases, RF would not consider variables that are not relevant. In addition, one of the objectives is to use modest computing power, which led us to choose random forest. A use of explainable AI for variable selection would have naturally extended to the tracking stage. The objective of this study is to provide some physical interpretation of the occurrence of a TC while having a frugal model (*i.e.,* few features). Last but not least, the number of variables is limited to those that are the most commonly available in climate models. This led us to use 20 variables, and we did not see the need to further reduce that number. Though we tested the sensitivity of the tracking by removing the least important variables, and showed in discussions that these are also important in controlling the false alarms.

[Figure]

Figure R1 – Evolution with respect to the number eliminated of features in abscissa (0 to 19) of POD, FAR, MCC, brier skill score BSS, area under the ROC curve (AUC) and the optimal threshold obtained from the ROC curve (TH) for a) ENP and b) NATL basins.

To further answer the reviewers' comment, we performed a recursive feature elimination (RFE) on the 6-fold cross-validation set-up (validation experiment) for **one RF**. It consists of eliminating the least important variable at each iteration. Figure R1 represents, with respect to the number of features eliminated in the abscissa (0 to 19), the evolution of POD, FAR, MCC, Brier skill score BSS, area under the ROC curve (AUC) and the optimal threshold obtained from the ROC curve (TH) for both basins. We see that the performance of RF remains quite stable until we remove 12 to 14 variables, which roughly corresponds to the number of variables we kept for the sensitivity test. The exception resides in FAR, which shows a 50% increase when we remove more than 6 variables, confirming the role of the least important in controlling the false alarms. This has been mentioned in lines 413-418 of the revised manuscript as :

**CHANGES–** To explore how many variables can be removed without deteriorating the performance of the tracker, a recursive feature elimination on the 6-fold cross-validation set-up (validation experiment) has been performed. It consists of eliminating the least important variable at each iteration. The results (not shown here) show that the performance of ERF remains stable until we remove 12 to 14 variables, which roughly corresponds to the number of variables we kept for the sensitivity test. The exception resides in FAR, which shows a 50% increase when we remove no more than 6 variables, confirming the role of the least important in controlling the false alarms.

✳*Comment–  Line 160 : Hyperparameter testing should be done for all RFs developed in this work and should be explicitly provided to readers for reproducibility. It is uncommon for RFs to be trained and validated without hyperparameter testing and tuning – I have to see any published works that used the default values given in whichever software package was being used. By testing and tuning the hyperparameters, the authors guarantee that the RFs are not "good by luck" and there is a repeatable process for future experimentation. Further, all hyperparameters should be listed in the manuscript/in a table.*

**RESPONSE–**  We realised a hyper-parameter tuning by performing a grid-search on three key parameters : the number of trees, - NTREE = 100, 250, 500 (default) -, the random number of features (predictors) considered to perform the best split - split try = 1, 2, 4 (default), 6, 10 - and the minimal size of end nodes - End node size = 1, 5, 10 (default), 20. The parameters' tuning is performed on the 6-fold cross-validation set-up (validation experiment) for **one RF**. Figure R2 represents the POD, FAR, MCC, BSS, AUC and for both basins and the 60 combination of parameters. We can see that the impact of the hyperparameters appears quite minimal. There is not one configuration that is the best for every index. The choice of the default parameters seems a reasonable choice. This question has been clarified in lines 156-160 of the revised manuscript as :

**CHANGES–**  A grid-search is performed on the three key parameters : (i) the number of trees, (ii) the random number of features considered to perform the best split to grow the trees and (iii) the minimal size of end nodes (not shown). Results showed that the impact of the hyperparameters is quite minimal, and no configuration of the hyperparameters yielded significantly better results. Therefore, the hyperparameters were set to the default values : 500 trees, 4 randomly chosen features to perform the best split and a minimal end node size of 10.

[Figure]

Figure R2 – Grid search results-In lines POD, FAR, MCC, Brier skill score BSS, area under the ROC curve (AUC) and the optimal threshold obtained from the ROC curve (TH) for a) ENP and b) NATL. In columns, are the results for different numbers of trees, and each 5 × 4 matrices are the results for a different random number of features considered to perform the best split (vertically) and different end node size (horizontally).

⁂*Comment– Line 225 : The authors should note some of the limitations of gini-based importance, namely the emphasis on input variables at the tops of the decision trees when proportions are largest. An alternative approach the authors could consider would be permutation importance, single-pass or multi-pass, which offers a more robust consideration of importance. See* McGovern *et al., 2019*

**RESPONSE–**  The permutation importance has been computed for **one RF** under the calibration experiment configuration (similar to Figure 9 of the manuscript). Figure R3 shows permutation importance for both basins. We see that some of the most important features are similar to those obtained from the Gini-based importance for both basins (RV850sd, MSLPmin). Otherwise, the order of the feature of importance between the two basins is very different. Besides, some variables like MSLPmax (ENP) or RV850min (NATL) are shown to be important, while UV10max is shown to be of lower importance (ENP). This makes the interpretation of feature importance unclear. Given these results, it seems that permutation-based importance is less consistent in our

case.

[Figure]

Figure R3 – Permutation-based importance.

✳*Comment– The authors use MCC, POD, and FAR based on a nebulous threshold of 50% to define TC track objects. Rather than using this approach, authors could leverage existing verification metrics that assess the probabilistic skill of the RF models (e.g., Brier Skill Score, Reliability diagrams). This approach would effectively assess the skill of the system at a range of probabilistic thresholds, which could also be leveraged for the area under the ROC statistics.*

[Figure]

Figure R4 – ROC curves.

**RESPONSE–** Thank you for this comment. We tested different thresholds below and above 50%. The effect was (i) that the higher the threshold, the lower the POD and (ii) that the lower the threshold, the higher the FAR. This behaviour was quasi-linear, so we chose the middle 50%. One can adapt this level according to the desired applications. In addition, when looking at BSS or AUC, they were extremely similar for different thresholds (as in Figures R1 and R2) and did not

help us to make a choice. We also looked into ROC curves to determine the optimal threshold. Figure R4 shows ROC curves for **one RF** under the validation experiment. For both basins, the optimal threshold is around 5%. With that threshold, we reach a POD of 100% but a FAR of 70%, which is fiercely undesired. This choice of 50% and the use of MCC, POD, and FAR was the best choice at hand to evaluate our tracker. The choice of threshold was clarified in lines 191-194 of the revised manuscript as :

**CHANGES–** Different thresholds below and above 0.5 have been tested (not shown). The result was (i) that the higher the threshold, the lower the ability to detect TC and (ii) that the lower the threshold, the higher the false alarms. This behaviour was quasi-linear, so we chose 0.5 to be performant to detect while having a low false alarm rate. One can adapt this level according to the desired applications.

**Technical Edits and Questions**

✳*Comment– Generally : the authors should spend a substantial amount of time proofreading the document for lingering grammar issues.*

**RESPONSE–** We tried as best as we could, we hope that the updated version of the manuscript is now satisfactory.

✳*Comment– Line 48 : Change to "this study focuses on data-driven algorithms using machine learning". Sometimes "so-called" can have a negative/inappropriate connotation, which I don't believe was your intent.*

**RESPONSE–** It has been removed and rewritten in line 47 of the revised manuscript

✳*Comment– Lines 93-96 : While I understand it is a long-held tradition to include a "table of contents paragraph" in this manner, you can remove this paragraph – it has no particular value for readers. The scientific structure of manuscripts has remained unchanged for decades, and every reader knows that methods will come next, results afterwards, and so on. If a reader is interested in a particular section, they can seek out the section header to know what is contained within.*

**RESPONSE–** Agreed and done

✳*Comment– Line 99 : Remove this single line*

**RESPONSE–** Done

✳*Comment– Line 99 : Remove "cyclonic" – seasons are not "cyclonic". Alternatively, can adjust to "cyclone seasons"*

**RESPONSE–** We modified accordingly in line 97 of the revised manuscript

✳*Comment– Line 106 : "Track records that do not provide"*

**RESPONSE–** Rewritten line 99-100 of the revised manuscript as :

**CHANGES–** Those labelled "spur", not providing maximum wind and minimum pressure, and not reaching the Tropical Storm (TS) stage, are removed.

✳*Comment– Lines 106-107 : If a TC undergoes extratropical transition, how is the transition from TC to extratropical TC handled ? Also, how is the TC's demise to the depression stage handled ? Only the TC achievement is mentioned here (i.e., genesis).*

**RESPONSE–** The transition from TC to extratropical TC is handled by limiting our study basin to 30°N. Thus, only TCs are considered and transitions to ET cyclones are not considered. For TC crossing this northward boundary, only the portion lying below 30°N is kept. In this section, we explain which TCs are kept for the following of the study. If the tropical storm level is not reached, the whole track is not considered. We do not handle the demise to the depression stage. Clarification about ET cyclones is given in lines 95-97 of the revised manuscript as :

**CHANGES–** First, extratropical cyclones are not considered in this study. Our study basins are limited to 30°N. Thus, only TCs are considered, and transitions to extratropical cyclones are not. For TC crossing this northward boundary, only the portion lying below 30°N is kept.

✳*Comment– Line 131 : Moisture is misspelt*

**RESPONSE–** Corrected in line 125 of the revised manuscript

✳*Comment– Lines 136-138 : The description here appears to have two statements in conflict with one another. First, the text says that every box has a vector of ones and zeros constructed : is this for every grid point in the box ? The next sentence says the box is encoded as a 1 or 0. Some additional clarity and perhaps the rewording of these sentences is needed to clarify the approach. I suspect it is the latter, but the wording is a bit confusing.*

**RESPONSE–** A vector of zeros and ones is constructed for each $20° \times 10°$ overlapping box not for each grid-point in the box. Lines 136-138 of the original manuscript are clarified in lines 129-131 of the revised manuscript as :

**CHANGES–** Then, for every box, a vector of zeros and ones is constructed  as follows : a box containing an IBTrACS point reaching TS intensity ($P_{\min} \leq 1005$ hPa and $u_{10} \geq 16 \mathrm{ms}^{-1}$) is coded 1, and 0 otherwise at every timestep.

✳*Comment– Line 134 : Why are the boxes not immediately adjacent to one another ? Could a TC be missed if it lies outside of the boxes in the white areas of Figure 1 ?*

**RESPONSE–** As indicated in line 134 of the original manuscript (line 130 in the revised version), the boxes overlap. Additionally, as stated in the caption of Figure 1, only every second box is shown to improve clarity. Plotting all overlapping boxes would make the figure unreadable.

✳*Comment– Lines 139-140 : What is the motivation for synthesising the ERA5 data in the boxes to single-statistic values ? Other works have used spatial regions to encode relevant spatial relationships into RFs [see Hill et al., 2020 ; Hill & Schumacher, 2021 ; Hill et al., 2023 ; Hill et al., 2024 ; Schumacher et al., 2021] and have had tremendous success, including deducing how those spatially oriented data contribute to forecast skill [Mazurek et al., 2025]. Others tackling severe weather hazards have taken a synthesising approach too [see Clark & Loken, 2022 ; Loken et al., 2022]. Were there any tests that also included the full box of ERA5 data to demonstrate that the single-value statistics were a better methodological choice ?*

**RESPONSE–** No formal test has been performed to demonstrate that the use of four single-value statistics instead of the whole field in the box was better. In this study, the TC tracking problem is handled as a binary classification problem. What we seek is the presence or absence of a TC within one box, given an atmospheric situation, regardless of its position. The position is deduced from the minimum of sea level pressure. Thus, four statistics summarising the spatial structure are preferred to describe the whole $20° \times 10°$ box. Furthermore, the ERA5 spatial resolution is $0.25°$, resulting in 3200 grid-points per box for each physical variables. Using 16000 predictors to predict one probability does not seem reasonable. Last but not least, as already stated in the last paragraph of the introduction, the ultimate goal of such a tracker is the tracking of TC in future climate simulations. Indeed, having many variables implies potential overfitting, impeded interpretation of the results and lower transferability to future climate simulations, in particular due to their systematic biases. Higher data frugality achieved by considering simple variable statistics instead of entire variable fields potentially improves the transferability of the tracking to climate simulations. This has been clarified in lines 139-143 of the revised manuscript as :

**CHANGES–** No formal test has been performed to demonstrate that using these four single-value statistics instead of the whole field in the box was better. Since only the presence or absence of a TC within one box, regardless of its position, is sought, these four statistics summarising the spatial structure are preferred to describe the whole $20° \times 10°$ box. Furthermore, the ERA5 spatial resolution is $0.25°$, resulting in 3200 grid-points per box for each physical variable. Using 16000 predictors to predict a single outcome does not seem reasonable.

✳*Comment– Lines 147-148 : This sentence is not needed – can be removed. All of this information is contained in the section headers.*

**RESPONSE–** Agreed, have been removed.

✳*Comment– Line 174-175 : To be consistent with both machine learning and atmospheric science literature, the "calibration" phase should be referred to as the "training" phase of the ERF. Then, you use cross-validation to validate the trained model on withheld periods – you don't use those withheld periods to "calibrate" the models.*

**RESPONSE–** As stated in lines 174-175 of the original manuscript : the calibration experiment consists in training ERF over the whole 1980-2021 period and validating over the whole period. Some clarifications have been made in lines 174-183 of the revised manuscript as :

**CHANGES–**

1. Calibration experiment : one  training of the ERF is made using the whole data during the 1980-2021 period and validated over the same period where all the tracks are sought to be reconstructed,

2. Validation experiment : a 6-fold cross-validation (see Fig. 2) where yellow years within each fold (35 years) are used to  train the ERF. The validation is performed over tracks reconstructed for all the validation years (in blue) from the six folds, allowing to validate ERF over the whole 1980-2021 period. This cross-validation is chosen to  minimise the effect of any potential trend and interannual variability in the TC statistics (frequency, intensity) and the changes in IBTrACS data quality. Most of the ERF evaluations will rely on this experiment.

3. Test experiment : from the  training performed over the whole  period in the calibration experiment for ENP (resp. NATL) basin, the TC tracks over the  NATL (resp. ENP) are reconstructed over the same period. This is done to evaluate the generalizability of ERF.

✳*Comment– Line 188 : Should RF actually be ERF?*

**RESPONSE–** Agreed and modified accordingly in lines 189 of the revised manuscript.

✳*Comment– Line 188 : Did you consider alternative probability thresholds (beyond just 50%) to assignment detected tracks (D)?*

**RESPONSE–** Thank you for that comment. We test different thresholds lower and above 50%. The effect was that the higher the threshold, the lower the POD and the lower the threshold, the higher the FAR. This behaviour was quasi-linear, so we chose the middle 50%. The choice of threshold was clarified in lines 191-194 of the revised manuscript as :

**CHANGES–** Different thresholds below and above 0.5 have been tested (not shown). The result was (i) that the higher the threshold, the lower the ability to detect TC and (ii) that the lower the threshold, the higher the false alarms. This behaviour was quasi-linear, so we chose 0.5 to be performant to detect while having a low false alarm rate. One can adapt this level according to the desired applications.

✳*Comment– Lines 251-253 : This text is best reserved for the figure caption – please move it there if not already. This text is just describing the figure, not the science.*

**RESPONSE–** Agreed and removed, already described in the caption.

✳*Comment– Figure 3 : It would be good to see the full distribution of MCC scores for the 100 RFs plotted as error bars, akin to a 95% confidence interval. Are the MCC values truly statistically indifferent? (It is hard to tell, but maybe this detail is plotted as light blue lines? If so, please try and make these lines clearer so they can be discerned and provide a description in the figure caption)*

**RESPONSE–** The boxplots for MCC have been made clearer, and we add a description of the boxplots in the caption of Figure 3 as follows :

**CHANGES–** The *top* and *bottom* fences are situated at 1.5 times the interquartile range from the box, and the dots are the values beyond these fences. The orange line represents the median

value.

*Comment–* *Lines 273-274 : What is meant by "calibration experiments"? Are you just evaluating the model's ability to detect storms over the testing period for which it was trained? It is to be expected that POD will be high and FAR low.*

**RESPONSE–** Yes, this is what we described as "calibration experiments". We hope the definitions of the different experiments in the revised manuscript, lines 174-183, make things clearer.

*Comment–* *Line 283-284 : Isn't a missed track by definition lower probability? Aren't hits/misses defined by probabilities greater than or less than 50%? These box plots in Figure 5b are being more or less constrained by the methods used, and don't necessarily provide much scientific reasoning for "FAs are less likely to happen than hits". The authors should reconsider the usefulness of this analysis regarding their methodological choices.*

**RESPONSE–** Thanks for these comments.

— Low probability means absence of TC. Missed tracks are tracks that were not detected in ERA5, although they were observed in IBTrACS. We explained in the manuscript that it is because they are absent in ERA5 and not a deficiency of ERF. This is something noteworthy to us.

— The 50 % threshold fixes if a TC is detected or not. Hit/FA/Miss/ assignment depends on whether a detected track was in IBTrACS or not (Hit/FA) and if an observed track was not detected (Miss). The method estimates the probabilities. These are evaluated conditionally on the Hit/FA/Miss/ assignment, which makes Fig. 5b interesting.

This has been clarified in lines 289-293 of the revised manuscript as :

**CHANGES–** Figure 5b  shows TC probabilities conditionally on its labelling as Hit, Miss, or FA. Probabilities associated with Hit tracks (median above 0.9) are substantially different compared to those associated  with FA (median a little above 0.6). This means that even if FA tracks are detected (probability >0.5) by ERF, FA are less likely to happen than Hits. Miss tracks are associated with very low probabilities, meaning they are completely missed by ERF while having been recorded in IBTrACS.

*Comment–* *Lines 320-322 : As mentioned earlier, they are also prescribed by the authors, so these results are not extremely surprising. See major comment above.*

**RESPONSE–** As stated in the answer to your first comment, we are surely prescribing the predictors, but we are not informing RF from which predictor it should learn to estimate the probability. So, analysing which are the most important predictors in the estimation of that probability is interesting.

*Comment–* *Lines 348-349 : This information is once again best reserved for the figure caption.*

**RESPONSE–** Assuming it refers to lines 318-319 (Figure 9), Agreed and removed, already described in the caption.

✳*Comment–* *Figure 10 : This is an excellent figure that clearly demonstrates how the RFs are learning the relevance of each predictor to drive the yes/no predictions.*
**RESPONSE–** Thanks for this comment.

**References**

Clark, Adam J. & Loken, Eric D. (**2022**). "Machine Learning–Derived Severe Weather Probabilities from a Warn-on-Forecast System". Weather and Forecasting. doi : `10.1175/WAF-D-22-0056.1`.

Hill, Aaron J., Herman, Gregory R. & Schumacher, Russ S. (**2020**). "Forecasting Severe Weather with Random Forests". Monthly Weather Review. doi : `10.1175/MWR-D-19-0344.1`.

Hill, Aaron J. & Schumacher, Russ S. (**2021**). "Forecasting Excessive Rainfall with Random Forests and a Deterministic Convection-Allowing Model". Weather and Forecasting. doi : `10.1175/WAF-D-21-0026.1`.

Hill, Aaron J., Schumacher, Russ S. & Green, Mitchell L. (**2024**). "Observation Definitions and Their Implications in Machine Learning–Based Predictions of Excessive Rainfall". Weather and Forecasting. doi : `10.1175/WAF-D-24-0033.1`.

Hill, Aaron J., Schumacher, Russ S. & Jirak, Israel L. (**2023**). "A New Paradigm for Medium-Range Severe Weather Forecasts : Probabilistic Random Forest–Based Predictions". Weather and Forecasting. doi : `10.1175/WAF-D-22-0143.1`.

Loken, Eric D., Clark, Adam J. & McGovern, Amy (**2022**). "Comparing and Interpreting Differently Designed Random Forests for Next-Day Severe Weather Hazard Prediction". Weather and Forecasting. doi : `10.1175/WAF-D-21-0138.1`.

Mazurek, Alexandra C., Hill, Aaron J., Schumacher, Russ S. & McDaniel, Hanna J. (**2025**). "Can Ingredients-Based Forecasting Be Learned ? Disentangling a Random Forest's Severe Weather Predictions". Weather and Forecasting. doi : `10.1175/WAF-D-23-0193.1`.

McGovern, Amy, Lagerquist, Ryan, Gagne, David John, Jergensen, G. Eli, Elmore, Kimberly L., Homeyer, Cameron R. & Smith, Travis (**2019**). "Making the Black Box More Transparent : Understanding the Physical Implications of Machine Learning". Bulletin of the American Meteorological Society. doi : `10.1175/BAMS-D-18-0195.1`.

Schumacher, Russ S., Hill, Aaron J., Klein, Mark, Nelson, James A., Erickson, Michael J., Trojniak, Sarah M. & Herman, Gregory R. (**2021**). "From Random Forests to Flood Forecasts : A Research to Operations Success Story". Bulletin of the American Meteorological Society. doi : `10.1175/BAMS-D-20-0186.1`.

---

## Author Comment (AC2)

"Ensemble Random Forest for Tropical Cyclone Tracking" by

P. Vaittinada Ayar et al.

We first would like to thank the anonymous reviewer for her/his thorough reading and very positive and constructive comments. We tried to take them into account as much as possible. A detailed point-by-point reply to these comments is provided below. Changes in the manuscript are indicated in **blue**.

**Answer to Referee #2**

**Summary :**

This study uses an ensemble of random forests (ERFs) to identify and track tropical cyclones (TCs) within ERA5 in the North Atlantic and East Pacific basins. The identified TCs and tracks are compared with observations taken from IBTrACS, and the ERF performance was compared with the Tempest Extreme tracking algorithm. Overall, the authors demonstrate that the ERF performs well in identifying and tracking observed TCs (high probability of detection and low false alarm ratio).

Beyond simply demonstrating that the ERF "works", the authors also nicely examined the characteristics of false alarms and misses. The authors found that missed TCs and false alarms were generally associated with short-duration storms that were of marginal tropical storm intensity. In addition, the authors examined which of the chosen predictors from ERA5 had the largest Gini-based feature importance and the contribution of each to the outcome of the random forest prediction using SHAP values.

I personally found the manuscript an interesting and useful application of ERFs. I particularly appreciated the authors' discussion on the misses, false alarms, and predictors that most informed the random forest outcome. I also believe the manuscript can be further improved through both the comments below and a more careful editing of the spelling and grammar within the text. I am specifically interested in encouraging the authors to more carefully consider the probability provided by the ERFs using traditional ensemble verification methods such as Brier Skill Score and ROC diagrams. It would also be of interest to better understand if the characteristics of the misses and false alarms from the ERF and Tempest Extreme exhibit any noteworthy differences in location, intensity, duration, or environmental conditions. Overall, I believe this is a study worthwhile of publication after addressing the below comments.

**Specific Comments :**

✳*Comment– One of the main benefits of the ERFs is the probabilities provided. I wish the authors examined this in more detail. I recommend that the authors reconsider the use of a strict threshold, greater than 50% probability, as defining a TC event. There is no requirement for this to be the cutoff, and the authors may wish to explore alternative thresholds. Furthermore, the authors may wish to examine the reliability of the ERFs by assessing whether the spread correctly represents the forecast uncertainty by examining the spread-error ratio. On average, the ensemble spread should be equal to the error. In addition, I suggest the authors examine reliability diagrams, which compare the forecasted probability with the observed frequency, ROC diagrams, and Brier skill score. Each of these analyses will help determine the benefits of the probability provided by the ERFs and may reveal weaknesses of the ensemble design.*

**RESPONSE–** Thank you for that comment. We tested different thresholds below and above 50%. The effect was that the higher the threshold, the lower the POD, while the lower the threshold, the higher the FAR. This behaviour was quasi-linear, so we chose the middle 50%. In addition, when looking at BSS or AUC, they were extremely similar for different thresholds and did not help us to make a choice. We also looked into ROC curves to determine the optimal threshold. Figure R1 shows ROC curves for **one RF** under the validation experiment. For both basins, the optimal threshold is around 5%. With that threshold, we reach a POD of 100% but a FAR of 70%, which is fiercely undesired. This choice of 50% and the use of MCC, POD, and FAR was the best choice at hand to evaluate our tracker. Concerning the spread-to-error ratio, we are not sure if it is pertinent for a binary (or any discrete) variable. Indeed, we want the probability to be a close as possible to 0 or 1, and the spread around 0 or 1 does not make sense. Though we computed it for ENP (0.044) and NATL (0.047). The choice of threshold was clarified in lines 191-194 of the revised manuscript as :

**CHANGES–** Different thresholds below and above 0.5 have been tested (not shown). The result was (i) that the higher the threshold, the lower the ability to detect TC and (ii) that the lower the threshold, the higher the false alarms. This behaviour was quasi-linear, so we chose 0.5 to be performant to detect while having a low false alarm rate. One can adapt this level according to the desired applications.

[Figure]

Figure R1 – ROC curves.

*✳Comment– I struggled to fully understand the details of the calibration, validation, and test experiments (L174-182). I am still a bit confused by the overlap between the calibration, validation, and testing periods. It appears from point 3 that the whole 1980-2021 period is used for testing. This is not a fair testing dataset, as the ERF was also tested using much of this same period. I believe the authors should perform testing using an entirely new period that was not used during training.*

**RESPONSE–** Eventhough the whole period is used for the evaluation, the validation experiment with the 6-fold cross-validation does not use the same data for training and validation, and the test experiment uses the whole period data from one basin for the training and data from the other

basin for the reconstruction, so data is never used in the training. Some clarifications have been made in lines 174-183 of the revised manuscript as :

**CHANGES–**

1. Calibration experiment : one  training of the ERF is made using the whole data during the 1980-2021 period and validated over the same period where all the tracks are sought to be reconstructed,

2. Validation experiment : a 6-fold cross-validation (see Fig. 2) where yellow years within each fold (35 years) are used to  train the ERF. The validation is performed over tracks reconstructed for all the validation years (in blue) from the six folds, allowing to validate ERF over the whole 1980-2021 period. This cross-validation is chosen to  minimise the effect of any potential trend and interannual variability in the TC statistics (frequency, intensity) and the changes in IBTrACS data quality. Most of the ERF evaluations will rely on this experiment.

3. Test experiment : from the  training performed over the whole  period in the calibration experiment for ENP (resp. NATL) basin, the TC tracks over the  NATL (resp. ENP) are reconstructed over the same period. This is done to evaluate the generalizability of ERF.

✳*Comment–  The authors also mentioned a potential change in the quality of IBTrACS with time in motivating their choice of validation data (every 6 years). I am curious if the authors tested how the performance of the ERF would change if they trained on an earlier period and then tested on a more recent period. This would be interesting for several reasons, including serving as an "easier" initial test for the potential application to future climate simulations, as the authors mention in the summary section.*

**RESPONSE–**  Thank you for that comment. The cross-validation scheme chosen in the study was only one possibility among others. We did a 6-fold test in which validation years were stacked (1980-1986,···,2015-2021). In the following table, we show the POD and FAR for tracks reconstructed with the probability obtained from one RF and the validation experiment using the "stacked" cross-validation : (i) over the whole 1980-2021 period (6-fold) and (ii) for the last fold (2015-2021 validation years). We see that the results for the whole period are similar to those of the validation experiment in Table 1 of the manuscript. The results are even better when the last fold, with the latest validation year, is considered (except the larger FAR for NATL, 10.7%).

| Validation | ENP | | NATL | |
|---|---|---|---|---|
| years | POD | FAR | POD | FAR |
| 1980-2021 | 78.1 | 8.4 | 77.5 | 8.5 |
| 2015-2021 | 85.6 | 4.7 | 87.0 | 10.7 |

This is now mentioned in lines 278-282 of the revised manuscript as :

**CHANGES–**  Note that the choice of validation data (one year every 6 years) in the cross-validation scheme in the study was only one possibility among others. A test (not shown), with a 6-fold cross-validation scheme in which validation years are stacked (1980-1986,···,2015-2021), yielded similar results for the validation experiment and, results are even better when only the last fold, with the latest validation years (2015-2021), is considered.

✳Comment– 300 km (L193) appears to be a generous threshold for the distance between an observed and identified TC to be considered a hit. This value is still probably small enough that it is identifying the same storm, but large enough that the center location may be off by the approximate size of the TC. Why was this value chosen and how sensitive are the results to this threshold?

RESPONSE– In Bourdin et al., 2022, a sensitivity analysis was conducted on this specific parameter in appendix D. In a nutshell, it was shown that this is not a sensitive parameter : choosing values between 200 and 400km did not make a difference in the final results. It was shown that the matching distance is usually below 100km for the SLP-based trackers (UZ & CNRM) – i.e. a couple of grid cells – and in more than 99% of the cases below 200km. Visual and statistical examination showed that, even with a large matching distance, these tracks are actual matches for at least a day. Taking values above 200km allows keeping the very few tracks where IBTrACS and the tracker disagree on part of the track, for example, the genesis in cases where two systems merged. For this reason, 300 km was selected as a reasonable value for the mean distance between the two tracks over the period for which they both exist. This has been clarified lines 197-199 of the revised manuscript as :

[Figure]

**Figure D3** Distribution of distance between matching detected and observed tracks. Whiskers display the 1st and 99th percentiles, and white points show the mean of the distributions. Outliers are not shown.

CHANGES– In Bourdin et al., 2022, a sensitivity analysis was conducted on the 300 km distance limit in appendix D. In a nutshell, it was shown that results are not sensitive to this limit, and 300 km was selected as a reasonable value.

✳Comment– It would be helpful to readers to define each of the predictors from ERA5 within a table in the supplementary material.

RESPONSE– The description of the variable is added as Table S1 of the supplementary material and is mentioned in line 118 of the revised manuscript as :

CHANGES– These variables are described in Table S1 of the supplementary material.

Table 1 – Variables description extracted from the Climate Data Store website. `https://cds.climate.copernicus.eu/datasets/`

| Variable | Abbreviation | Unit | Description |
|---|---|---|---|
| Mean sea level pressure | MSLP | Pa | This parameter is the pressure (force per unit area) of the atmosphere at the surface of the Earth, adjusted to the height of mean sea level. It is a measure of the weight that all the air in a column vertically above a point on the Earth's surface would have if the point were located at mean sea level. It is calculated over all surfaces - land, sea and inland water. |
| 10-m wind intensity | UV10 | $m\ s^{-1}$ | This parameter can be calculated by combining the eastward (U) and northward (V) components of the 10m wind. U is the horizontal speed of air moving towards the east, and V is the horizontal speed of air moving towards the north, at a height of ten metres above the surface of the Earth. |
| Total column water vapour | TCWV | $kg\ m^{-2}$ | This parameter is the total amount of water vapour in a column extending from the surface of the Earth to the top of the atmosphere. This parameter represents the area-averaged value for a grid box. |
| Relative Vorticity at 850 hPa pressure level | RV850 | $s^{-1}$ | This parameter is a measure of the rotation of air in the horizontal, around a vertical axis, relative to a fixed point on the surface of the Earth. On the scale of weather systems, troughs (weather features that can include rain) are associated with anticlockwise rotation (in the Northern Hemisphere), and ridges (weather features that bring light or still winds) are associated with clockwise rotation. It is extracted at the 850 hPa pressure level. |
| Geopotential thickness between 500 and 300hPa pressure levels | THZ300_Z500 | m | This parameter is obtained from the gravitational potential energy of a unit mass, at a particular location, relative to mean sea level. It is also the amount of work that would have to be done, against the force of gravity, to lift a unit mass to that location from mean sea level. The geopotential height can be calculated by dividing the geopotential by the Earth's gravitational acceleration, g (=9.80665 m $s^{-2}$), and the thickness is the difference in height between two levels of pressure. |

✳*Comment–  I am interested in understanding if the characteristics of the false alarms and misses are similar to the ERFs and Tempest Extremes. I suggest the authors recreate Figures 5, 6, and 7 for Tempest Extreme within the supplemental figures. This analysis may help identify the strengths and weaknesses of each approach.*

**RESPONSE–**  Figures 5 to 7 have been reproduced for UZ and both basins. They have been added to the supplementary material as Figures S7 to S12. Panel b of Figure 5 and the two top panels of Figure 7, showing the probability associated with the tracks, are irrelevant for UZ. For both basins, the track properties obtained from UZ are similar to those obtained with ERF (see Figs. S7, S8, S10 and S11). The main difference resides in the properties of missed tracks. These tracks are similarly short but more frequent and slightly more intense (higher maximum wind and lower SLP). Properties of FAs are similar for UZ and ERF in the NATL basin. In the ENP basin, FAs from UZ are significantly more frequent and intense than the FAs generated by ERF. The spatial distribution of Miss and FA of the UZ tracker is also similar to that of ERF (Fig. S9 and S12). Miss tracks are distributed over the entire domain with low intensity, and FA tracks are mostly located in the primary development region and coastal areas, where TCs are typically weaker.

These comments have been added in lines 363 to 371 of the revised manuscript as :

**CHANGES–**  Figures 5 to 7 have been reproduced for UZ and both basins. They have been added to the supplementary material as Figures S7 to S12. Panel b of Figure 5 and the two top panels of Figure 7, showing the probability associated with the tracks, are irrelevant for UZ. For both basins, the track properties obtained from UZ are similar to those obtained with ERF (see Figs. S7, S8, S10 and S11). The main difference resides in the properties of missed tracks. These tracks are similarly short but more frequent and slightly more intense (higher maximum wind and lower

SLP). Properties of FAs are similar for UZ and ERF in the NATL basin. In the ENP basin, FAs from UZ are significantly more frequent and intense than the FAs generated by ERF. The spatial distribution of Miss and FA of the UZ tracker is also similar to that of ERF (Fig. S9 and S12). Miss tracks are distributed over the entire domain with low intensity, and FA tracks are mostly located in the primary development region and coastal areas, where TCs are typically weaker.

**Technical edits :**

✳*Comment– L2 : change "evanesce" to "weaken"*

**RESPONSE–** Done

✳*Comment– L37-38 : Another tracking algorithm the authors may be interested in referencing here is TRACK (Hodges, 1994). This algorithm differs from others in that it is more general and tracks all vorticity maximums and only later filters out TCs using a warm core threshold. Hodges, K. I., 1994 : A General Method for Tracking Analysis and Its Application to Meteorological Data. Mon. Wea. Rev., 122, 2573–2586, https ://doi.org/10.1175/1520-0493(1994)122<2573 :AGMFTA>2.0.CO ;2.*

**RESPONSE–** We are aware of the TRACK algorithm. A comparison of TRACK, UZ (TempestExtremes) and two other tracking algorithms was performed previously in [Bourdin et al., 2022]. For this study, we wanted to keep it simple and compare to only one "traditional" tracker. We chose TempestExtremes/UZ for two reasons : the RF tracker was targeted towards IBTrACS as a ground truth, and the comparison showed that TempestExtremes/UZ had the lowest False Alarm Rates with respect to IBTrACS, and TempestExtremes is faster and easier to run than TRACK.

We agree that TRACK is designed in a way that allows it to capture a wide range of features. Unfortunately, that is also the reason why it has a large number of false alarms. In our opinion, TRACK's strength lies in its ability to capture the early and late stages of TCs. In the case of the present study, we are focusing on the mature stage of TCs as captured within IBTrACS. It should also be noted that, in a similar way to TRACK, TempestExtremes is a software that can be used to implement many algorithms, and the warm core criterion can be removed to detect all depressions (or vortices if vorticity is used as a variable).

✳*Comment– L144 : The authors should more carefully describe what is meant by "standardized". This is important for reproducibility.*

**RESPONSE–** This has been clarified in lines 137-138 of the revised manuscript as :

**CHANGES–** standardised (*i.e.* centred and divided by the standard deviation).

✳*Comment– L252 : I suggest the authors replace "different subsampling of zeros" with language more physically intuitive.*

**RESPONSE–** Line 251-253 of the original manuscript was removed since it is a repetition of the caption of figure 3. "different subsampling of zeros" has been replaced by "different subsampling of non-TC situations (*i.e.* zeros)"

✳*Comment–* *Figure 6 : The layout of the figure panels in Figure 6 are a bit confusing. I was repeatedly confusing panels (a) and (b). I suggest revising the layout to avoid this.*

**RESPONSE–** We invert panel labels a) to b).

✳*Comment–* *L311 : remove "basin"*

**RESPONSE–** Done, now in line 320 of the revised manuscript.

✳*Comment–* *L314-315 : What is the basis for this hypothesis?*

**RESPONSE–** IBTrACS record relies on the human interpretation of satellite images from different meteorological centres. Since these TCs have intensities around the tropical storm threshold ($P_{\min} \leq 1005$ hPa and $u_{10} \geq 16\mathrm{ms}^{-1}$), some of them can be recorded as not having reaching the tropical storm intensity and were therefore excluded from our study.

✳*Comment–* *L323 : Change "with" to "which".*

**RESPONSE–** Done, now in line 331 of the revised manuscript.

✳*Comment–* *L325 : Change "since they are associated with the strong surface winds and the location of the cyclone eye, respectively".*

**RESPONSE–** Done, now in line 332 of the revised manuscript.

✳*Comment–* *L332-333 : A transition would be helpful here to emphasize the different information provided by each of these analyses.*

**RESPONSE–** Agreed. Some clarifications have been made in lines 340-341 of the revised manuscript as :

**CHANGES–** Indeed, we want to evaluate the ability of RFs to learn the relevance of each predictor to drive each TC/non-TC prediction.

✳*Comment–* *L362-362 : I suggest splitting this into two sentences. Ending the first sentence after "literature".*

**RESPONSE–** Done, now line 380-381 of the revised manuscript.

✳*Comment–* *L390-391 : The end of this sentence, "indicating us to be…" should be revised.*

**RESPONSE–** Done, now in line 409 of the revised manuscript and modified as :

**CHANGES–** indicating  that we ought to be cautious when removing predictors.

**References**

Bourdin, S., Fromang, S., Dulac, W., Cattiaux, J. & Chauvin, F. (**2022**). "Intercomparison of four algorithms for detecting tropical cyclones using ERA5". Geoscientific Model Development. doi : `10.5194/gmd-15-6759-2022`.

---

## Author Response (AR2)

**"Ensemble Random Forest for Tropical Cyclone Tracking" by**

**P. Vaittinada Ayar et al.**

We first would like to thank the anonymous reviewer for her/his thorough reading and very positive and constructive comments. We tried to take them into account as much as possible. A detailed point-by-point reply to these comments is provided below. Changes in the manuscript are indicated in **blue**.

**Answer to Referee #1**

**Overview**

The authors addressed a number of my previous comments in their responses, as well as edits made to the manuscript. I also thank the authors for taking the time to edit and proofread the manuscript for grammatical issues.

**Remaining Comments**

\*Comment— 1. Please be consistent with your use of acronyms. For example, you use "TCs" in Line 122 to refer to Tropical Cyclones, yet "TC" in Line 127 also refers to Tropical Cyclones. However, the "TC" acronym is defined on line 117 as "Tropical Cyclones" — so should the acronym be TC or TCs in this case? Please edit the text as you see appropriate to have consistent plural/singular use of this (and other) acronyms.

**RESPONSE** Agreed, and modified throughout the paper.

\*Comment— For the record, standardisation is not necessary for random forest applications (lines 153-154).

**RESPONSE**— Thank you for that comment with which we agree. The standardisation has been done to anticipate tracking TCs in climate models with biases compared to ERA5. The standardisation removes part of the mean and variance biases of the climate models and potentially eases the transferability to the tracker to climate models without recalibration. This has been clarified in the paper in lines 138-141 of the revised manuscript as:

**CHANGES**— Note that standardisation is not necessary for the current application of random forests. However, it has been made here to anticipate tracking TCs in climate models with biases compared to ERA5. The standardisation removes part of the mean and variance biases of the climate models and potentially eases the transferability to the tracker to climate models without recalibration.

\*Comment— 3. Lines 201-204: As long as testing is done for a different basin, i.e., model trained on ENP for the full period and "tested" on NATL for the same period, this approach should be valid. However, if an

ENP model is trained on a period and subsequently tested on that same period, I would have significant concerns about misrepresenting skill.

**RESPONSE**— As mentioned in the manuscript, three types of experiments are set. Assuming that the reviewer refers to the calibration experiment in the second part of this comment [*However...*]: The calibration experiment is only performed as a first-order evaluation of the tracker. In machine learning, the first step is to evaluate the ability of the model to reproduce the training data. The calibration experiment has that sole purpose. As mentioned in the paper, the major part of the evaluation is performed through the validation and test experiments. It has been clarified in lines 179-180 of the revised manuscript as:

**CHANGES**— It is only performed as a first-order evaluation of the tracker and its ability to reproduce the training data.

\*Comment— 4. Line 193: Why are there 100 Random Forests used for each subsampling test? The Random Forest by nature is an ensemble of Decision Trees – why do you need 100 RFs?

**RESPONSE**— The subset of zeros provided to each RF is different. The effect of subsampling zeros on the track reconstruction is evaluated with 100 RFs. We showed the effect of the subsample to be marginal in Figure 3 of the manuscript. It has been clarified in line 176 of the revised manuscript as:

**CHANGES— with a different subset of zeros provided to each RF**

\*Comment— 5. Lines 210-214 and previous review comment: Both I and another reviewer asked to see forecast skill as a function of probability, e.g., reliability diagrams. I'm not convinced that the 50% threshold used here is not more-or-less arbitrary. Since you have developed a probabilistic prediction system, probabilistic skill metrics should be computed to evaluate skill properly and robustly.

**RESPONSE–** The 50% threshold choice has not been chosen arbitrarily, but was based upon a key practical criterion: having the ability to track TC (high POD) while having a low FAR. As mentioned in the answer to your previous comments, (i) the higher the threshold, the lower the POD and (ii) the lower the threshold, the higher the FAR. This behaviour was quasi-linear, so we chose the middle 50%. We also looked at BSS or AUC, but they did not help us choose a threshold: they were extremely similar for different thresholds due to the unbalanced nature of the classification. We also looked into ROC curves to determine the optimal threshold. For both basins, the optimal threshold is around 5%. With that threshold, we reach a POD of 100% associated with a 70% FAR, which is obviously undesired. Similarly, we are not convinced that a reliability diagram is adapted to define a TC threshold, especially for such an unbalanced problem. Figure R1 shows these diagrams for ERF under the validation experiment and both basins. From these figures, we are not able to define a threshold. The 50% threshold seems a good practical compromise.

Figure R1 – Reliability diagramme curves for both basin and the histogram associated with the prediction (bottom). The histograms are cut to 3000; low probability values reach maximum frequencies above  $5 \times 10^5$ .